# Thymosin β4 Regulates the Differentiation of Thymocytes by Controlling the Cytoskeletal Rearrangement and Mitochondrial Transfer of Thymus Epithelial Cells

**DOI:** 10.3390/ijms25021088

**Published:** 2024-01-16

**Authors:** Yuyuan Ying, Nana Tao, Fengjie Zhang, Xunuo Wen, Meiru Zhou, Jianli Gao

**Affiliations:** School of Pharmaceutical Sciences, Zhejiang Chinese Medical University, Hangzhou 310053, China; 202111124011129@zcmu.edu.cn (Y.Y.); 202111124011114@zcmu.edu.cn (N.T.); 202321114011093@zcmu.edu.cn (F.Z.); 202321124011160@zcmu.edu.cn (X.W.); 202321124011185@zcmu.edu.cn (M.Z.)

**Keywords:** immune, thymus, Thymosin β4, thymus epithelial cells, thymocytes, actin, differentiation, cytoskeletal, mitochondria, aging

## Abstract

The thymus is one of the most crucial immunological organs, undergoing visible age-related shrinkage. Thymic epithelial cells (TECs) play a vital role in maintaining the normal function of the thymus, and their degeneration is the primary cause of age-induced thymic devolution. Thymosin β4 (Tβ4) serves as a significant important G-actin sequestering peptide. The objective of this study was to explore whether Tβ4 influences thymocyte differentiation by regulating the cytoskeletal rearrangement and mitochondrial transfer of TECs. A combination of H&E staining, immunofluorescence, transmission electron microscopy, RT-qPCR, flow cytometry, cytoskeletal immunolabeling, and mitochondrial immunolabeling were employed to observe the effects of Tβ4 on TECs’ skeleton rearrangement, mitochondrial transfer, and thymocyte differentiation. The study revealed that the Tβ4 primarily regulates the formation of microfilaments and the mitochondrial transfer of TECs, along with the formation and maturation of double-negative cells (CD4^−^CD8^−^) and CD4 single-positive cells (CD3^+^TCRβ^+^CD4^+^CD8^−^) thymocytes. This study suggests that Tβ4 plays a crucial role in thymocyte differentiation by influencing the cytoskeletal rearrangement and mitochondrial transfer of TECs. These effects may be associated with Tβ4’s impact on the aggregation of F-actin. This finding opens up new avenues for research in the field of immune aging.

## 1. Introduction

Aging and aging-related chronic disorders stand out as significant contributors to global mortality [1]. The decline in the normal function of the immune system associated with aging is encapsulated by the term “immunosenescence” [2]. The thymus, a primary lymphoid organ necessary for T lymphocyte development, coordinates adaptive immune response and holds a pivotal role in immunosenescence [3]. The structural and functional degradation of the thymus is widely recognized as a primary factor contributing to immunosenescence [4]. Consequently, international research in aging and geriatrics has focused on addressing the question of “How to inhibit thymus atrophy?” Despite numerous research achievements, there is currently a lack of significant breakthroughs in health medicine and longevity research.

The thymus is located in the mediastinum, behind the sternum. It is composed of two identical lobes. Each lobe is divided into a central medulla and a peripheral cortex [5]. The cortex, mainly consisting of dense thymocytes (i.e., thymic lymphocytes, precursor cells of T cells) and cortical thymic epithelial cells (cTECs), does not completely enclose the medulla, with adjacent medullary lobules interconnected. Thymocytes in the cortex originate from hematopoietic stem cells, undergoing proliferation and differentiation into small lymphocytes from the shallow to deep layers. In the medulla, intramedullary lymphocytes are sparse, while medullary thymic epithelial cells (mTECs) are abundant, scattered around thymic corpuscles [3,6,7]. T lymphocytes’ development and selection are not cell-autonomous processes, requiring continuous input from various cells within thymic microenvironments [8]. Chemokines secreted by stromal cells, including cTECs and mTECs, guide thymocytes through the thymus from the medullary to cortical regions and back before their egress to the periphery [3]. With ageing, the formation of lipid-loaded cells in the thymus increases, and adipose tissue replaces functional cortical and medullary areas, contributing to a decline in thymus function [9,10]. In this intricate process, TECs emerge as one of the most critical stromal cell types in the thymus for the production of functional thymocytes, both quantitatively and phenotypically [11]. cTECs regulate T lymphocytes’ commitment and positive selection, while mTECs provide mechanisms for central tolerance formation in these T lymphocytes [12], which is key for maintaining immune system balance. Recently, it has been demonstrated that the intricate three-dimensional (3D) network of TECs incorporates muscle-like contractions involving actin and myosin [13,14,15].

Thymosin, a family of biologically active molecules with hormone-like properties, was initially isolated from the calf thymus by Goldstein and White in 1966 [16,17]. Comprising three peptide families—α, β, and γ [18]—thymosin exhibits high similarity in the amino acid sequences of its various polypeptide components [19]. Thymosin β family, including Tβ4, 10, and 15, plays a pivotal role as regulatory factors in the structure of TECs. Tβ4, the most abundant member of the thymosin β family, is encoded by the *Tmsb4x* gene and ubiquitously present in almost all cells and tissues, except red blood cells, across vertebrates and invertebrates. It is a hydrophilic multifunctional peptide with an N-terminal serine acetylation, possessing a molecular weight of 5 kDa, 43 amino acids, and a pI of 5.1 [20].

In 1991, Safer et al. identified Tβ4 as a G-actin sequestering peptide [21]. Tβ4 binds to actin monomers in a ratio of 1:1 through a spatial mechanism, inhibiting actin monomer polymerization. This inhibition makes it challenging to reach the critical concentration required for polymerization and actively participates in regulating actin’s polymerization and depolymerization, thereby influencing cell activity [22]. Tβ4 has been found to promote the migration of mouse breast cancer cells in vitro and enhance the metastasis of mouse breast cancer in vivo [23]. In bone development, Tβ4 demonstrates anti-apoptotic effects, accelerates the cell cycle, and promotes osteoblast proliferation [24]. Overexpression of *Tβ4* in human colon cancer cells may induce epithelial–mesenchymal transition (EMT) by upregulating recombinant integrin-linked kinase (ILK) [25]. Tβ4 potentially regulates VEGF and MMP-2 levels through the Wnt/β-catenin/LEF-1 signaling pathway, impacting blood vessel growth and activating cell migration. Additionally, Tβ4 exhibits a potent effect on hair growth and hair follicle development [26], along with anti-inflammatory properties that prevent the release of injury-induced pro-inflammatory cytokines and chemokines [27,28]. However, the mechanism by which Tβ4 maintains the form and function of TECs remains unclear. Therefore, this study focuses on exploring the impact of Tβ4 on the structure and spatial distribution of TECs, aiming to initially reveal the potential mechanisms of Tβ4 regulation on the differentiation and development of thymocytes.

## 2. Results

### 2.1. Expression of Tβ4 in Different Treated iTECs

To confirm the successful construction and functionality of stable cells, adenovirus, and small interfering RNA (siRNA), six groups of iTECs were employed, comprising Normal (Normal iTECs), LVCon (iTECs infected with LVCON077 control lentivirus [LV] labeled with green fluorescent protein [GFP]), LVshTβ4 (iTECs infected with LV-Tmsb4x-RNA interference [RNAi] lentivirus labeled with GFP), AdCon (iTECs infected with pENTER control adenovirus), AdTMSB4X (iTECs infected with TMSB4X *Tβ4*-overexpressed adenovirus), NC siRNA (iTECs treated with control siRNA), and Tβ4 siRNA (iTECs treated with Tβ4 siRNA). These groups were cultured with a growth medium. Cells were observed and photographed under bright fields and green fluorescence excitation (Figure 1a). The findings revealed that the lentivirus transfection efficiency for the LVCon group was 81.97%, and for the LVshTβ4 group, it was 94.87% (Figure 1b). This demonstrated the successful transfection of LVCON077 and LV-Tmsb4x-RNAi lentiviruses into iTECs. Subsequently, Western Blot was employed to assess Tβ4 levels in each group. The results indicated no significant differences between the Normal, LVCon, AdCon, and NC siRNA groups. In comparison with the LVCon group, the Tβ4 levels in the LVshTβ4 group significantly decreased by 39.76%. Conversely, compared with the AdCon group, the Tβ4 levels in the AdTMSB4X group significantly increased by 23.74%. Furthermore, compared with the NC siRNA group, the Tβ4 levels in the Tβ4 siRNA group significantly decreased by 58.95% (Figure 1c,d). Additionally, these results were further validated by RT-qPCR (Figure 1e). Taken together, lentivirus-stable cells were successfully constructed, and adenovirus and siRNA effectively operated in iTECs.

### 2.2. Tβ4 Has No Significant Effect on iTEC Proliferation

Our previous research results findings indicated that Tβ4 did not exert a significant effect on the proliferation of 4T1 cells [29]. Consistent with these results, crystal violet staining demonstrated no significant differences in cell proliferation at 24 h, 48 h, and 72 h among the Normal, LVCon, LVshTβ4, AdCon, and AdTMSB4X groups (Figure 1d,e). Consequently, it can be concluded that Tβ4 does not have a significant effect on iTEC proliferation.

### 2.3. Tβ4 Maintains Normal Thymus Morphology by Inhibiting Cortical Thymocyte Overflow

To investigate the impact of Tβ4 on the thymus, we harvested cell supernatant from the Normal, LVCon, LVshTβ4, AdCon, and AdTMSB4X groups. Subsequently, we cultured 2-week-old mice thymus with the collected cell supernatant for a duration of 7 days. Thymocyte overflow was recorded and photographed under a microscope on day 7. The results revealed a significant increase of 139.17% in thymocyte overflow in the LVshTβ4 group compared to the LVCon group (Figure 2a,b). Moreover, Stereographic photographs of the thymus were captured on day 1 and day 7, illustrating a significant reduction of 6.49% in the maximum cross-section of the thymus in the LVshTβ4 group compared to the LVCon group (Figure 2c,d). Thymus weight reduction analysis further demonstrated a significant increase of 24.15% in the LVshTβ4 group compared to the LVCon group (Figure 2e).

Subsequently, histopathological examination using H&E staining revealed that in comparison with the LVCon group, the boundary between the cortex and medulla of the thymus was indistinct (Figure 2f). Statistical analysis indicated a significant decrease of 45.82% in cortical thymocyte numbers. However, there was no significant difference in medullary thymocyte numbers among all groups (Figure 2g). These findings suggest that Tβ4 may play a role in maintaining the normal morphology and size of the thymus by influencing the overflow of cortical thymocytes.

### 2.4. Tβ4 Affects the Differentiation and Spatial Distribution of TECs

Cytokeratin 5 (CK5) and cytokeratin 8 (CK8) are members of the cytokeratin family, In the thymus, the expression of CK5 and CK8 is mainly related to the proliferation, differentiation, and function of TECs. Cytokeratin 8 (CK8) and cytokeratin 5 (CK5) are markers of TECs in the thymic cortex and medulla [30,31]. To investigate the effect of Tβ4 on the reticular structure of TECs, we employed CK5 and CK8 to label the thymus. The results showed that, despite a reduction in thymocytes in the LVshTβ4 group, the reticular structure of TECs remained evident. However, in the AdTMSB4X group, both cortical and medullary TECs exhibited a less reticular and more stellate appearance (Figure 3a). According to the above results, we can preliminarily speculate that the number of thymocytes in the LVshTβ4 group is reduced, but TECs still have a relatively complete reticular structure, which may still provide certain support and regulatory effects. In the thymus of the AdTMSB4X group, the reticular structure of TECs decreased and still increased, which may lead to abnormal development and function of thymocytes.

To gain further insights into the potential role of Tβ4 in thymocyte maturity and migration, we examined the mRNA levels of key factors involved in these processes within thymic tissue. Notably, vascular cell adhesion molecule-1 (VCAM-1), intercellular cell adhesion molecule-1 (ICAM-1), L-selectin (CD62L), SQUAMOSA promoter binding protein-like (SPL) transcription factor, and sphingosine1-phosphate receptor 1 (S1P1) were analyzed, with S1P1 and CD62L being crucial factors mediating cell migration [32,33]. RT-qPCR analysis revealed that compared with the LVCon group, the LVshTβ4 group exhibited significantly decreased mRNA levels of *VCAM-1* and *ICAM-1*, while *CD62L* and *S1P1* levels were significantly increased. Conversely, compared with the AdCon group, the AdTMSB4X group displayed significantly increased mRNA levels of *VCAM-1* and *ICAM-1*, with a simultaneous decrease in *CD62L* and *S1P1* levels (Figure 3b). It can be preliminarily speculated that the decrease of VCAM-1 and ICAM-1 in the thymus of the LVshTβ4 group may lead to the decrease of adhesion and interaction between thymocytes and TECs, while the increase of CD62L and S1P1 may enhance the migration and localization ability of thymocytes. The AdTMSB4X group was the opposite.

The reticular structure of TEC is vital for the development and selection of thymocytes in the thymus. This intricate network is not solely formed by TECs themselves but rather through interactions between TECs, other cell types, and the extracellular matrix. To assess the impact of Tβ4 on cell interactions in the thymus, we injected 10 μL lentivirus or adenovirus into the chest from the center of the first pair of ribs. Thymus organs were then removed on the 7th-day post-injection and observed using a transmission electron microscope. The results revealed that thymocytes in the LVCon group, AdCon, and AdTMSB4X groups displayed smooth cell membranes. In contrast, the cell membrane surface of the LVshTβ4 group exhibited small protrusions between cells (Figure 3c). Regarding mitochondrial morphology, the LVCon group, AdCon, and AdTMSB4X groups predominantly displayed a short stick shape, while the LVshTβ4 group exhibited mostly spherical mitochondria (Figure 3d).

In summary, our study demonstrated that Tβ4 may influence adhesion function between thymocytes, regulate and support the reticular structure of TECs, affect mitochondrial function, and thus affect the migration ability of thymocytes.

### 2.5. Tβ4 Affects the Differentiation and Development of Thymocytes

CD4 and CD8 are markers of T cell subsets. Mature T cells have specific T cell receptors (TCR) and surface markers, such as CD3. The differentiation and maturation of thymocytes will go through four stages: double-negative (DN) cells, double-positive (DP) cells, single-positive cells, and mature cells [34,35]. To assess whether Tβ4 affects thymocyte differentiation and development, we conducted a flow analysis of thymocyte subsets from thymus cultured in vitro for 7 days. The results revealed that Tβ4 influenced the differentiation of CD4^−^CD8^−^, CD4^+^CD8^+^, and CD3^+^TCRβ^+^CD4^+^CD8^−^. In comparison with the LVCon group, the LVshTβ4 group exhibited a significant decrease of 28.70% in the proportion of CD4^−^CD8^−^, a significant increase of 16.68% in the proportion of CD4^+^CD8^+^, and a significant decrease of 21.90% in the proportion of CD3^+^TCRβ^+^CD4^+^CD8^−^. Conversely, compared with the AdCon group, the AdTMSB4X group showed a significant increase of 23.72% in the proportion of CD4^−^CD8^−^, a decrease of 5.38% in the proportion of CD4^+^CD8^+^, and a significant increase of 41.62% in the proportion of CD3^+^TCRβ^+^CD4^+^CD8^−^ (Figure 4a). The statistical results are presented in Figure 4b.

Subsequently, we isolated thymocytes and cultured them for 24 h and 48 h, respectively. The results from the isolated thymocytes were consistent with the above findings from the cultured thymus. After 24 h culture, significant differences were observed in CD4^−^CD8^−^, CD4^+^CD8^+^, CD3^+^TCRβ^+^, and CD3^+^TCRβ^+^CD4^+^CD8^−^ subsets. In comparison with the LVCon group, the LVshTβ4 group exhibited a significant decrease in the proportion of CD4^−^CD8^−^, CD3^+^TCRβ^+^, and CD3^+^TCRβ^+^CD4^+^CD8^−^ by 15.18%, 3.91%, and 15.88%, respectively, while the proportion of CD4^+^CD8^+^ was significantly increased by 5.77%. To the contrary, compared with the AdCon group, the AdTMSB4X group showed a significant increase in the proportion of CD4^−^CD8^−^, CD3^+^TCRβ^+^, and CD3^+^TCRβ^+^CD4^+^CD8^−^ by 21.16%, 2.76%, and 16.08%, respectively, while the proportion of CD4^+^CD8^+^ was significantly decreased by 7.56% (Figure 4c). The statistical results are depicted in Figure 4d.

When thymocytes were cultured for 48 h, significant differences were observed in CD4^−^CD8^−^, CD4^+^CD8^+^, and CD3^+^TCRβ^+^CD4^+^CD8^−^ subsets. In comparison with the LVCon group, the LVshTβ4 group exhibited a significant decrease in the proportion of CD4^−^CD8^−^ and CD3^+^TCRβ^+^ by 21.47% and 12.71%, respectively, while the proportion of CD4^+^CD8^+^ was significantly increased by 5.77%. Conversely, compared with the AdCon group, the AdTMSB4X group showed a significant increase in the proportion of CD4^−^CD8^−^ and CD3^+^TCRβ^+^ by 33.97% and 29.59%, respectively, while the proportion of CD4^+^CD8^+^ was significantly decreased by 8.56% (Figure 4e). The statistical results are presented in Figure 4f.

In summary, our study demonstrated that Tβ4 influences the differentiation and development of thymocytes primarily by affecting CD4^−^CD8^−^ and CD3^+^TCRβ^+^CD4^+^CD8^−^ subsets. After *Tβ4*-knockdown, it promoted the development of DN T cells to DP T cells but inhibited the differentiation of mature cells. *Tβ4*-overexpression was the opposite.

### 2.6. Tβ4 Controls the Cytoskeletal Rearrangement and Mitochondrial Transfer of TECs

To investigate the impact of Tβ4 on the cytoskeleton of TECs, we initially fluorescently labeled G-actin and F-actin in TECs treated with adenovirus and siRNA. The laser scanning confocal microscope was then utilized to observe the complete skeleton structure, microfilament formation, and adhesive spots of TECs. The results demonstrated that, in comparison with the NC siRNA group, the Tβ4 siRNA group exhibited continuous microfilaments and dense adhesive spots. Additionally, when compared with the AdCon group, the AdTMSB4X group displayed short and poorly continuous microfilaments (Figure 5a). Further analysis of fluorescence intensity for F-actin and G-actin revealed that, relative to the NC siRNA group, the Tβ4 siRNA group showed a significant increase of 17.57% in F-actin fluorescence intensity and a significant decrease of 63.93% in G-actin fluorescence intensity. Conversely, compared with the AdCon group, the AdTMSB4X group exhibited a significant decrease of 28.95% in F-actin fluorescence intensity and a significant increase of 81.85% in G-actin fluorescence intensity (Figure 5b).

Subsequently, we investigated the levels of cytoskeleton-related proteins, including PROFILIN-2, CYCLIND1, α-TUBLIN, VIMENTIN, and E-CADHERIN, through Western Blot (Figure 5c). The results demonstrated that, in comparison with the LVCon group, the LVshTβ4 group exhibited a significant decrease in PROFILIN-2 and E-CADHERIN levels of 43.57% and 50.06%, respectively (Figure 5d,e). These findings were further validated by RT-qPCR, which showed a consistent reduction in the mRNA levels of *profilin-2* and *e-cadherin* in the LVshTβ4 group, with a decrease of 55.77% and 50.35%, respectively (Figure 5f,g). Sosne et al. emphasized that Tβ4 stimulated in vitro conjunctival epithelial cell migration and resulted in altered focal adhesion formation [36]. Aleksandra et al.’s research indicated that Tβ4 was not only a component of focal adhesions (FAs) and interacted with several FAs components but also regulated FAs formation [37]. Therefore, when *Tβ4* is knocked down, the adhesion ability of cells to the matrix might be affected, resulting in the reduction of cell adhesion and migration. PROFILIN-2 and E-CADHERIN were proteins associated with cell adhesion and migration [38,39], and their decline might be the result of decreased cell adhesion and migration ability. When *Tβ4* was over-expressioned, it might lead to excessive enhancement of cell adhesion and migration, resulting in abnormal cell behavior. The decline in PROFILIN-2 and E-CADHERIN might be the result of excessive enhancement of cell adhesion and migration, and cells might exhibit abnormal adhesion and migration behavior.

In a previous study, we observed differences in the mitochondrial morphology of thymocytes in various treatment groups (Figure 3d). The statistical analysis of mitochondrial numbers revealed a significant increase of 74.04% in the LVshTβ4 group compared to the LVCon group, while the AdTMSB4X group exhibited a significant decrease of 50.00% compared to the AdCon group (Figure 3e). To delve into the cause of this phenomenon, we labeled the mitochondria and nuclei of thymocytes and iTECs, respectively, and co-cultured them. After co-culturing, F-actin was labeled. The results illustrated that relative to the normal group, TECs in the Tβ4 siRNA group displayed a long synaptic structure connected with thymocytes, revealing visible mitochondrial transmission in the synaptic structure with significant mitochondrial staining in thymocytes. In contrast, the synaptic structure of TECs in the 0.1 ng/μL Tβ4 group was less apparent, and the mitochondrial staining was minimal in thymocytes (Figure 5h).

In conclusion, our study demonstrated that Tβ4 regulates the polymerization and depolymerization of actin, influencing cytoskeletal rearrangement in TECs and mitochondrial transfer between TECs and thymocytes.

## 3. Discussion

The two forms of G-actin/F-actin undergo conversion under specific physiological conditions, a phenomenon, known as “Tread Milling”, which occurs concurrently in the positive and negative microfilament assembly [22]. The ratio of G-actin/F-actin is maintained in balance within the cell and plays a role in various physiological functions. Changes in the cell’s G-actin/F-actin ratio impact cell morphology during migration and homeostasis. Actin, in terms of cell structure, assembles into filaments to create a cytoskeleton that shapes and stabilizes cells [40]. In terms of cell movement, actin is involved in myodynamic processes, dynamically regulating assembly and deaggregation for cellular motion. The cytoskeleton’s remodeling and dynamic changes enable cells to stretch, contract, move, and deform, forming protrusions like platypodia and filamentous pseudopodia, establishing adhesion, and retracting tails [41,42].

Tβ4, functioning as an actin-sequestering protein, plays a crucial role in regulating the G-actin/F-actin balance. Our study demonstrated the multilayered role of Tβ4 in TECs.

Scheller et al. discovered that *Tβ4*-deficient platelets exhibit reduced G-actin levels, increased F-actin levels, accelerated fibrinogen diffusion, and clot contraction. Interestingly, our findings align with theirs. *Tβ4*-knockdown iTECs displayed continuous microfilaments and dense adhesive spots, while *Tβ4*-overexpression iTECs had short and discontinuous microfilaments. Fluorescence intensity statistics indicated higher F-actin fluorescence and weaker G-actin fluorescence in *Tβ4*-knockdown iTECs, with the opposite observed in *Tβ4*-overexpression iTECs. Profilins, small actin-binding proteins, compete directly with Tβ4 for actin binding [43,44]. Western Blot results revealed a significant decrease in PROFILIN-2 level in *Tβ4*-knockdown iTECs. Additionally, the E-CADHERIN level significantly decreased in *Tβ4*-knockdown iTECs, consistent with Krishnanand et al.’s findings [45].

Transmission electron microscopy results showed that *Tβ4*-knockdown led to irregular cell membranes in thymocytes and an increased number of mitochondria. while *Tβ4*-overexpression resulted in smoother cell membranes and reduced mitochondrial numbers. Co-culture of thymocytes and TECs indicated that *Tβ4*-knockdown TECs were more likely to form protruding or tubular structures and promote mitochondrial transfer, whereas culturing with 0.1 ng/μL Tβ4 yielded opposite results. Furthermore, *Tβ4*-knockdown correlated with a significant decrease in cortical thymocytes, suggesting that Tβ4 regulates thymocyte differentiation and movement to maintain the normal thymus structure by influencing TECs’ microfilament formation and mitochondrial transfer.

To comprehend the molecular mechanisms underlying these changes, it is crucial to consider the role of key molecules in cell adhesion and migration. Swaminathan et al. demonstrated that the retrograde flow of F-actin cytoskeleton aligns ICAM-1 bound integrins at the leading edge, reinforcing cellular adhesions [46]. Heidi et al.’s research indicated that β2-integrins and the β2-integrin ligand ICAM-1 can repress dendritic cell-mediated thymocytes’ activation and Th17 differentiation [47]. Additionally, Aleksandar et al. emphasized the role of CD62L in regulating monocyte protrusion during transendothelial migration, supported by mounting evidence in the literature [19]. Audrey et al. showed that S1P directs the exit of thymocytes from lymph nodes, where thymocytes are initially activated, into lymph [48]. Our results indicated that, upon *Tβ4* knockdown, the mRNA levels of *VCAM-1* and *ICAM-1* significantly decreased, while the mRNA levels of *CD62L* and *S1P1* increased. Conversely, *Tβ4* overexpression yielded reversed results. Therefore, our study demonstrated that Tβ4 expression influenced *VCAM-1*, *ICAM-1*, *CD62L*, and *S1P1* levels, thereby affecting adhesion connections between thymocytes. Immunofluorescence results revealed a distinct reticular structure in TECs of the LVshTβ4 group, whereas cortical and medullary TECs in the AdTMSB4X group exhibited less reticular and more stellate features. Flow cytometry analysis further revealed that Tβ4 influenced thymocyte differentiation and development, primarily impacting double-negative cell (CD4^−^CD8^−^) and CD4 single-positive cell (CD3^+^TCRβ^+^CD4^+^CD8^−^) subsets. Consequently, Tβ4 plays a crucial role in the spatial recombination of TECs and guides thymocyte development by regulating crosstalk between TECs and thymocytes, ultimately affecting the differentiation and development of the latter.

In summary, our study demonstrated that Tβ4 regulates the polymerization and depolymerization of actin, influencing cytoskeletal rearrangement and mitochondrial transfer in TECs. This, in turn, affects the motility and development of thymocytes, ultimately preserving the normal structure and function of the thymus. While many questions remain to be addressed to fully understand the mechanism of Tβ4 in the thymus, our study provides important insights, suggesting Tβ4 as a potential new target for delaying thymus immune aging. Tβ4 may represent a breakthrough in the effort to maintain the normal structure and function of the thymus and delay aging. Ongoing research will further investigate the mechanisms underlying cytoskeletal rearrangement and mitochondrial transfer in delaying thymus aging.

## 4. Materials and Methods

### 4.1. Animals and Cells

Two 8-week-old SPF BALB/c female mice and two 8-week-old SPF BALB/c male mice were purchased from Hang Zhou Hangsi Biotec Co., Ltd. (Hangzhou, China) and raised at the Experimental Animal Research Center of Zhejiang Chinese Medicine University. The institute was accredited by the Association for Assessment and Accreditation of Laboratory Animal Care (AAALAC). All animal experiments were carried out in accordance with the National Institutes of Health guide for the care and use of Laboratory animals. The ethics approval number for the animals used in this study was IACUC-20230522-13. BALB/c female mice and male mice freely mated, and 67 newborn mice were raised to 2 weeks old for in vitro or in vivo experiments.

Mouse-immortalized thymus epithelial cells (iTECs) were constructed in our published paper [29]. The cells were cultured in Dulbecco’s Modified Eagle Medium (C11995500BT; Gibco, Tokyo, Japan) containing 10% fetal bovine serum (11011-8611; EVERY GREEN, Nagano, Japan) and 1% penicillin-streptomycin solution (CR-15140; Cienry, Huzhou, China) in a 37 °C CO_2_ incubator (Forma 310; Thermo Fisher Scientific, Waltham, MA, USA) with 5% CO_2_ and saturated humidity. When the cell density reached 80–90%, trypsinization was performed and cell passage was performed at 1:3.

### 4.2. Establishment of Lentivirus Stable Cells

*Tβ4*-knockdown lentivirus (LV-Tmsb4x-RNAi (92194-11)) was inserted into lentivirus vector GV493 (Shanghai GeneChem Co., Ltd. (Shanghai, China)), target sequence was 5′-TAAGTCGAAGTTGAAGAAA-3′. Lentivirus was used to infect the iTECs, appropriate amount of HitransG virus infection agent (Shanghai GeneChem Co., Ltd. (Shanghai, China)) was introduced into the culture medium to enhance the infection efficiency. The infection efficiency was observed using an inverted fluorescence microscope (TH4-200; OLYMPUS). When the fluorescence rate reached 80% or above, 3 μg/mL puromycin (A610593; Sangon Biotech, Shanghai, China) was added for 48 h to screen drug-resistant iTECs. mRNA level of *Tβ4* was assessed by RT-qPCR. Two stable cell lines, iTECs infected with control lentivirus (iTECs-LVCon) and iTECs infected with *Tβ4*-knockdown lentivirus (iTECs-LVshTβ4) were obtained.

### 4.3. Preparation and Transfection of Adenovirus

NM_021109 was cloned into a pAD-Amp vector and subsequently packaged with adenovirus to generate the *Tβ4*-overexpressed adenovirus (TMSB4X, Vigene Biosciences, Rockville, MD, USA). The control counterpart utilized was the pENTER control adenovirus (Vigene Biosciences). The denovirus was transfected into iTECs following the manufacturer’s provided instructions, and 0.4 mg/mL polybrene (C0351; Beyotime, Nantong, China) was introduced into the culture medium to enhance transfection efficiency. RT-qPCR was employed to assess the mRNA level of *Tβ4* and verify the efficiency of overexpression.

### 4.4. Crystal Violet Staining

After replacing the medium, the cells were fixed with 4% paraformaldehyde for 15 min at 24 h, 48 h, and 72 h. Subsequently, 0.2% crystal violet dye was applied, left for 20 min, washed, and dried. Images were captured under a bright field using the inverted fluorescence microscope (TH4-200; OLYMPUS, Tokyo, Japan). Following this, 33% acetic acid was added, and the plate orifice was shaken using an orbital shaker (TS-200; Kylin-Bell, Nantong, China) for 10 min. The optical density values were then measured at a wavelength of 570 nm using the multifunctional microplate detector (Synergy H1; BioTek, Winooski, VT, USA).

### 4.5. Thymus Culture In Vitro

#### 4.5.1. Collection of Cell Supernatants

Cells, including iTECs, iTECs-LVCon, and iTECs-LVshTβ4, were cultured for 24 h. Control adenovirus and *Tβ4*-overexpressed adenovirus were added to normal iTECs, respectively. The groups were labeled as Normal, LVCon, LVshTβ4, AdCon, and AdTMSB4X. After being infected for 24 h, the original culture medium was discarded, cleaned once with PBS, and 1 mL of new culture medium was added. The cell supernatant was collected at 24 h, 48 h, and 72 h, and mixed together for tissue culture.

#### 4.5.2. Collection and Cultivation of Thymus Tissue

Mice were sacrificed for thymus collection. After the surface connective tissue was removed, the thymus was washed three times with PBS containing 1% penicillin-streptomycin solution and placed into cell supernatants collected from 2.5.1. All of the thymus were cultured in a constant temperature incubator at 37 °C with 5% CO_2_ and saturated humidity for 7 days. The morphological changes of thymus organs were recorded using a stereo microscope (Zeiss, Jena, Germany).

### 4.6. Thymocytes Co-Culture with iTECs

Cells, encompassing iTECs, iTECs-LVCon, and iTECs-LVshTβ4, were cultured for 24 h. Control adenovirus and *Tβ4*-overexpressed adenovirus were individually introduced to iTECs, respectively. The designated groups were labeled as Normal, LVCon, LVshTβ4, AdCon, and AdTMSB4X. Thymocytes were procured by grinding and screening the thymus from mice. The treated thymocytes were co-cultured with iTECs for 24 h and 48 h.

### 4.7. H&E Staining

For histological analysis, the cultured thymus was obtained from 2.5.2. Then, it was fixed in 4% formalin, embedded in paraffin blocks, and cut into 4 μm sections. Subsequently, sections were stained with hematoxylin-eosin dye (purchased from Nan Jing Jiancheng Technology Co., Ltd. (Nanjing, China)), and images were collected using a microscope (MF43-N; Mshot, Guangzhou, China).

### 4.8. Pleural Injection of Lentivirus/Adenovirus and Transmission Electron Microscope of Thymus Tissues

Fifteen female two-week-old BALB/c mice were divided into five groups in a randomized method: Normal, LVCon, LVshTβ4, AdCon, and AdTMSB4X. For each mouse, 10 μL lentivirus or adenovirus was injected into the chest from the center of the first pair of ribs about 5 mm deep with a 1 mL syringe. On the 7th day after injection, the thymus tissue was collected and trimmed into 1 mm^3^ tissue blocks with blades and fixed in 2.5% glutaraldehyde. The samples were prepared at the Public Platform of Pharmaceutical Research Center, Academy of Chinese Medical Sciences, Zhejiang Chinese Medical University. The images were collected by a transmission electron microscope (H-7650; HITACHI, Tokyo, Japan).

### 4.9. Immunofluorescence of CK5 and CK8 in Thymus

After formalin fixation, the cultured thymus was dehydrated, paraffin-embedded tissues were cut into 4 μm slices, dewaxed, and placed in EDTA antigen repair solution (purchased from Bei Jing Zhongshan Jinqiao Biotec Co., Ltd. (Beijing, China)) for antigen repair. The primary antibodies used were as follows: Rat anti-Mouse Cytokeratin 5 (CK5) antibody (dilution 1:200, BF0493; Affinity Biosciences, Cincinnati, OH, USA) and Rabbit anti-Mouse Cytokeratin 8 (CK8) antibody (dilution 1:200, ab53280; Abcam, Cambridge, UK). The second antibodies were Mouse anti-Rabbit IgG-FITC (dilution 1: 200, sc-2359; Santa Cruz, Santa Cruz, CA, USA) and F[ab’]2 of Goat Anti-Mouse IgG (H+L) (dilution 1:200, DW-GAR7512; Hang Zhou Dawen Biotec Co., Ltd. (Hangzhou, China)). Seal with Antifade Mounting Medium with DAPI (P0131; Beyotime, Shanghai, China). Photographs were taken using a fluorescence microscope (AXIO SCOPE.A1; Zeiss).

### 4.10. Total RNA Extraction and RT-qPCR

The thymus tissues obtained from 2.5.2 were ground with a high-throughput tissue grinder (SCIENTZ-48) and total RNA was extracted according to the instructions of Tissue RNA Purification Kit Plus (RN002plus; ES Science, Shanghai, China). The total RNA of cells was extracted according to the protocol of the RNA-Quick Purification Kit (RN001; ES Science, Shanghai, China). The purity and concentration of RNA were determined with UV-Vis Spectrophotometer (Monad, Suzhou, China). According to the Fast All-in-One RT Kit (with gDNA Remover) (RT001; ES Science, Shanghai, China) protocol, a corresponding program was set on the A200 Gradient Thermal cycler (LongGene, Hangzhou, China) to reverse transcribe total RNA into DNA. Real-time PCR was performed using 2×SG Fast qPCR Mix (B639271; Sangon Biotech, Shanghai, China). mRNA levels were normalized to GAPDH mRNA levels using the 2^−ΔΔCt^ method. All sequence information was provided in Appendix A.

### 4.11. Antibodies and Flow Cytometry

The thymus tissues obtained from 2.5.2 were ground and sieved through 70 µm filters (BS-70-XBS; Biosharp, Hefei, China). FITC Hamster Anti-Mouse CD3e (BD553061), APC Rat Anti-Mouse CD4 (BD553051), PE Rat Anti-Mouse CD8a (BD553032), and BB700 Hamster Anti-Mouse TCRβ (BD745846) were adopted to stain isolated thymocytes. The flow cytometry (CytoFlex S, Beckman) collected all flow data and analyzed the result using CytExpert 4.1.

Collect thymocytes obtained from 4.6. FITC Hamster Anti-Mouse CD3e (BD553061), APC Rat Anti-Mouse CD4 (BD553051), PE Rat Anti-Mouse CD8a (BD553032) and BB700 Hamster Anti-Mouse TCRβ (BD745846) were adopted to stain thymocytes. The flow cytometry (CytoFlex S; Beckman, Brea, CA, USA) collected all flow data and analyzed the result using CytExpert 4.1.

### 4.12. Immunolabelling of F-actin and G-actin in iTECs

The iTECs were inoculated on 9 cm-diameter round coverslips (BS-09-RC; Biosharp) and treated with NC siRNA (sc-37007; Santa Cruz) or Tβ4 siRNA (sc-45217, Santa Cruz), and Lipo6000^TM^ transfection reagent (C0526FT; Beyotime) according to the manufacturer’s protocol for 24 h at 37 °C. Cell fixation fluid with 4% paraformaldehyde was used to fix the iTECs. Simultaneous visualization of F-actin and G-actin was realized by F-actin-specific Oregon Green^TM^ 488 phalloidin (working concentration 1.65 μM, O7466; Thermo Fisher Scientific) and G-actin–specific Deoxyribonuclease I, Alexa Fluor^TM^ 594 (working concentration 9 μg/mL, D12372; Thermo Fisher Scientific) in iTECs. The cell nucleus was stained with DAPI. Tissue slices were sealed with Antifade Mounting Medium. Images were acquired by Laser scanning confocal microscope (LSM880; Zeiss).

### 4.13. Western Blot

Cells were washed with 4 °C pre-cooled PBS and lyzed with RIPA buffer (high) (R0010; Solarbio, Beijing, China) (containing 1 mM PMSF (P0100; Solarbio)). Total protein concentration was detected with the BCA Protein Assay Kit (P0012; Beyotime). SDS-PAGE Sample Loading Buffer (5X) (P0015L; Beyotime) denatured the protein. The following primary antibodies were used: β-Actin (13E5) Rabbit mAb (dilution 1:1000, 4970S; CST), Vimentin (D21H3) XP^®^ Rabbit mAb (dilution 1: 1000, 5741T; CST), Cyclin D1 antibody (A-12) (dilution 1:1000, sc-8396; Santa Cruz), anti-alpha tubulin antibody (dilution 1: 5000, ab52866; abcam), E-cadherin rabbit polyclonal antibody (dilution 1:500, 208741-AP; proteintech), Profilin-2 antibody (4K-6) (dilution 1:1000, sc-100955; Santa Cruz), and Human Thymosin beta 4 Antibody (dilution 1:200, AF6796; R&D). The secondary antibodies were: Anti-rabbit IgG, HRP-linked Antibody (dilution 1:2000, 7074P2; CST), HRP-conjugated Rabbit anti-mouse IgG (dilution 1: 5000, D110098; Santa Cruz), and Rabbit anti-sheep IgG(H+L) (dilution 1:3000, ab6747; abcam). The PVDF membrane (Bio-RAD, Hercules, CA, USA) was sealed with NON-Fat Powdered Milk (A600669-0250; BBI Life Sciences, Shanghai, China) and incubated with an ECL chemiluminescent substrate kit (Supersensitive) (BL523A; Biosharp). Two-color prestain protein marker 10~250 kDa (WJ102; Shanghai Epizyme Biomedical Technology Co., Ltd., Shanghai, China) and Biotinylated Molecular Weight Markers (MW001; RD) were used for protein localization. The membrane was developed by Cheniluminescence Imaging System (GD50202; Monad, Suzhou, China) and quantified by Image J v1.53e.

### 4.14. Transfer of Mitochondria between TECs and Thymocytes

Before co-culturing, the iTECs of the siRNA group underwent a 6 h treatment with siRNA. Subsequently, iTECs and thymocytes were individually labeled with 0.1 μM MitoBright LT Red (MT11; Dojindo, Kumamoto, Japan) and 0.1 μM MitoBright LT deep Red (MT12; Dojindo) for 20 min at 37 °C. Following this, cells were stained with Hoechst 33258 (C1017; Beyotime) for 10 min at 37 °C. The co-culturing process lasted for 18 h, during which the 0.1 ng/μL Tβ4 group was co-cultured with 0.1 ng/μL Thymosin β4 (#3390; Tocris, Bristol, UK). After co-culturing, cells were fixed with 4% paraformaldehyde for 5 min. F-actin-specific Oregon Green^TM^ 488 phalloidin (working concentration 1.65 Μm, O7466; Thermo Fisher Scientific) was then applied and left for 35 min. Images were captured using a Laser scanning confocal microscope (LSM880; Zeiss).

### 4.15. Statistical Analysis

If not otherwise specified, all results are expressed as mean ± SEM. All statistics analyses were calculated by performing unpaired Student’s *t*-test, one-way ANOVA test with Tukey posttest, or two-way ANOVA analysis with Tukey posttest using GraphPad Prism 9.5.1. *p*-values < 0.05 were considered as significant differences.

## Figures and Tables

**Figure 1 ijms-25-01088-f001:**
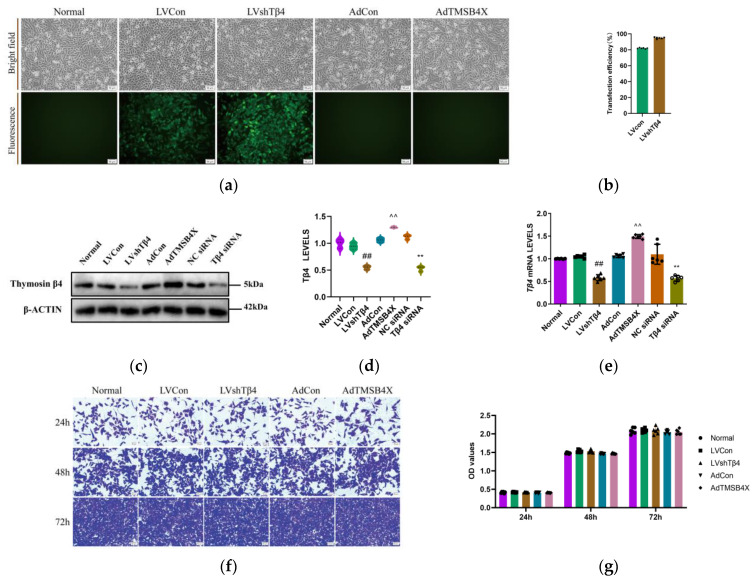
Analysis of Tβ4 levels and cell proliferation capacity. (**a**) Representative bright field and fluorescence images of cells from the Normal, LVCon, LVshTβ4, AdCon, and AdTMSB4X groups. Scale bar = 50 μm; (**b**) Lentivirus transfection efficiency data for LVCon and LVshTβ4 groups; (**c**) Representative Western Blot images of β-ACTIN, Thymosin β4; (**d**) Statistical results of Tβ4 levels. Protein levels were normalized using the corresponding β-ACTIN level; (**e**) *Tβ4* mRNA levels in Normal, LVCon, LVshTβ4, AdCon, AdTMSB4X, NC siRNA, and Tβ4 siRNA groups. mRNA levels were normalized to *GAPDH* mRNA levels using the 2^−ΔΔCt^ method. (**f**) Representative images of crystal violet staining at 24 h, 48 h, and 72 h for Normal, LVCon, LVshTβ4, AdCon, and AdTMSB4X groups. Scale bar = 50 μm; (**g**) Crystal violet staining results at 24 h, 48 h, and 72 h for Normal, LVCon, LVshTβ4, AdCon, and AdTMSB4X groups. One-way ordinary ANOVA tests were used in (**b**,**d**,**e**), and a two-way ANOVA test was used in (**g**). Data for (**b**,**d**,**e**,**g**) are presented as mean ± SEM from at least three independent experiments. Statistical significance was determined by comparing with the LVCon group (^##^
*p* < 0.01), AdCon group (^^^^
*p* < 0.01), and NC siRNA group (** *p* < 0.01).

**Figure 2 ijms-25-01088-f002:**
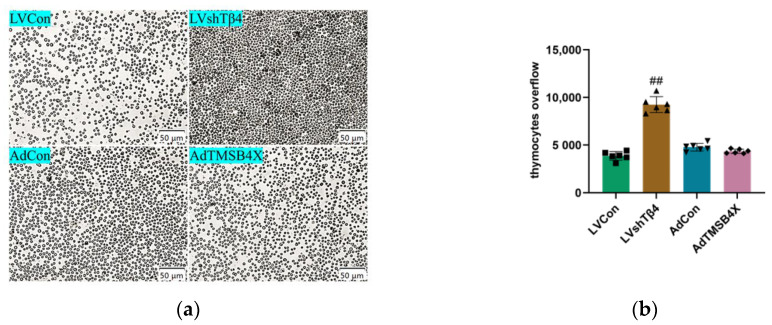
Impact of Tβ4 on thymus morphology, structure, and cell number. (**a**) Representative images of thymocytes overflow in LVCon, LVshTβ4, AdCon, and AdTMSB4X groups. Scale bar = 50 μm; (**b**) Statistical results of (**a**); (**c**) Representative images of the thymus captured with a stereo microscope in LVCon, LVshTβ4, AdCon, and AdTMSB4X groups. Scale bar = 250 μm; (**d**) Maximum cross-section reduction of thymus (%) in LVCon, LVshTβ4, AdCon, and AdTMSB4X groups; (**e**) Wight reduction of thymus (%) in LVCon, LVshTβ4, AdCon, and AdTMSB4X groups; (**f**) H&E representative images of thymus in LVCon, LVshTβ4, AdCon, and AdTMSB4X groups. Scale bar = 250 μm (upper), 50 μm (lower); (**g**) The counting area for the lower image in (**f**). Statistical results of cortical and medullary thymocyte numbers in LVCon, LVshTβ4, AdCon, and AdTMSB4X groups. One-way ordinary ANOVA tests were used in (**b**,**d**,**e**), and a two-way ANOVA test was used in (**g**). Each symbol represents a separate data point. Data for (**b**,**d**,**e**,**g**) are presented as mean ± SEM from at least three independent experiments. Statistical significance was determined by comparing it with the LVCon group (^##^
*p* < 0.01).

**Figure 3 ijms-25-01088-f003:**
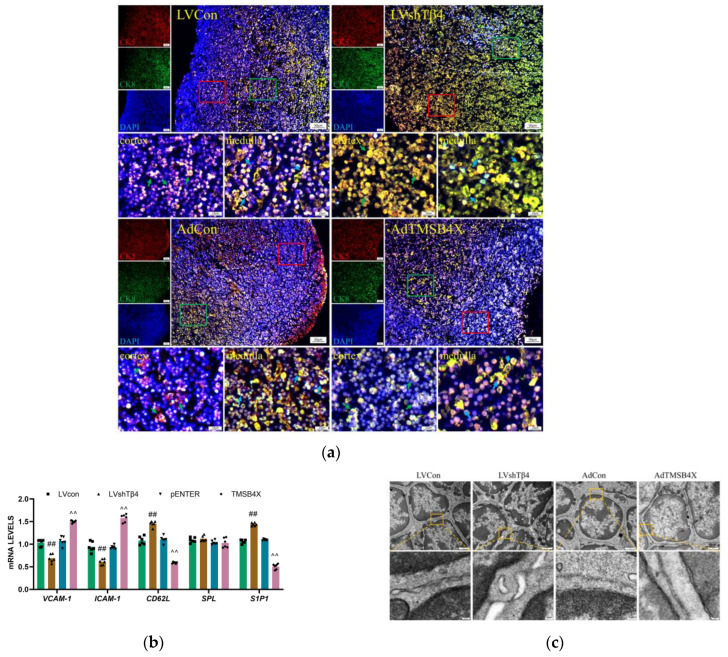
TEC differentiation and spatial distribution. (**a**) Representative immunofluorescence images of CK5 (red) and CK8 (green) in the 7-day cultured thymus, with nuclei stained using DAPI (blue). Dotted lines outline cortical and medullary regions. Representative immunofluorescence images of enlarged selected areas in the red and green boxes, green arrows indicate cTECs, and blue arrows indicate mTECs. Scale bar = 50 μm (upper), 10 μm (lower); (**b**) Statistical result of mRNA levels of *VCAM-1*, *ICAM-1*, *CD62L*, *SPL*, and *S1P1* in the thymus detected by qPCR. mRNA levels were normalized using *GAPDH* by the 2^−ΔΔCt^ method; (**c**) Representative images of thymocytes captured by transmission electron microscope and enlarged images of the selected areas in the orange boxes. Scale bar = 1 μm (bigger), 100 nm (smaller); (**d**) Representative images of thymocytes captured by transmission electron microscope with enlarged images of selected areas in the orange boxes. Blue arrows indicate mitochondria. Scale bar = 1 μm (bigger), 100 nm (smaller); (**e**) The counting area for the larger image in (**d**) and the statistical results of the number of mitochondria. A one-way ordinary ANOVA test was used in (**e**), and a two-way ANOVA test was used in (**b**). Each symbol represents a separate data point. Data for (**e**,**b**) are presented as mean ± SEM from at least three independent experiments. Statistical significance was determined by comparing the LVCon group (^##^
*p* < 0.01) and the AdCon group (^^^
*p* < 0.05, ^^^^
*p* < 0.01).

**Figure 4 ijms-25-01088-f004:**
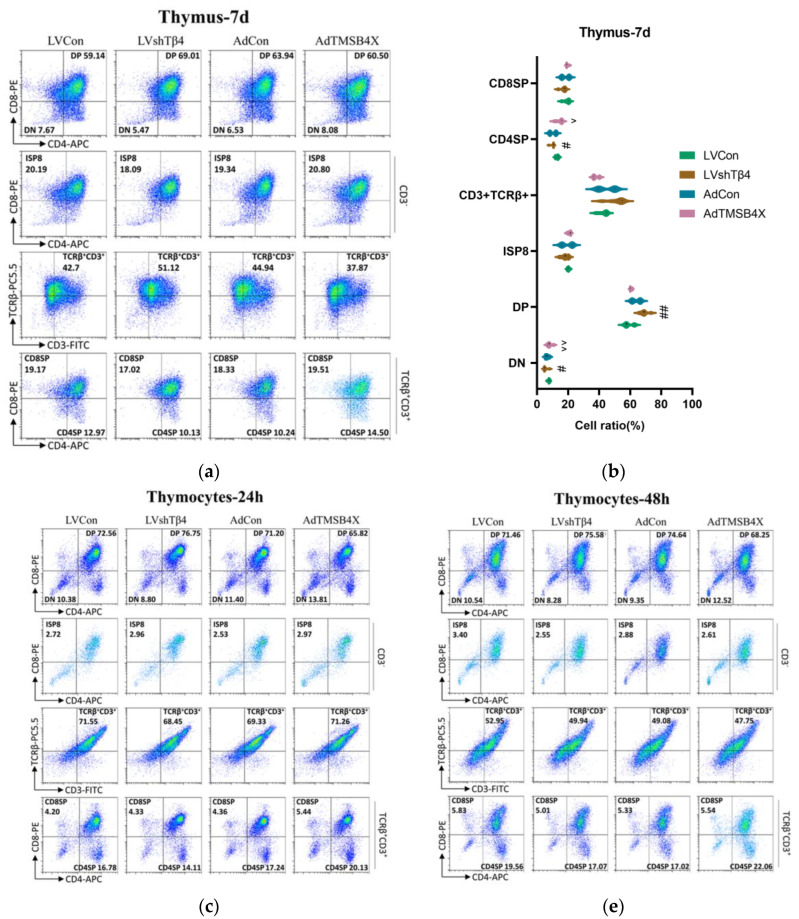
Thymocyte differentiation and development. (**a**) Flow cytometry results depicting thymocyte differentiation from thymus cultured for 7 days in vitro; (**b**) Statistical results of (**a**); (**c**) Flow cytometry results showing thymocytes cultured for 24 h in vitro; (**d**) Statistical results of (**c**); (**e**) Flow cytometry results illustrating thymocytes cultured for 48 h in vitro; (**f**) Statistical results of (**e**). Percentage frequencies of double-negative (DN) cells (CD4^−^CD8^−^), double-positive (DP) cells (CD4^+^CD8^+^), immature single-positive CD8^+^ (ISP8) cells (CD3^−^CD4^−^CD8^+^), and TCRβ^+^CD3^+^ in total cells. Percentage of mature single-positive CD8^+^ (CD8SP) cells (CD3^+^TCRβ^+^CD4^−^CD8^+^) and mature single-positive CD4^+^ (CD4SP) cells (CD3^+^TCRβ^+^CD4^+^CD8^−^) in thymus TCRβ^+^CD3^+^ cells. A two-way ANOVA test was used in (**b**,**d**,**f**). Data for (**b**,**d**,**f**) are presented as mean ± SEM from at least three independent experiments. Statistical significance was determined by comparing the LVCon group (^#^
*p* < 0.05, ^##^
*p* < 0.01) and the AdCon group (^^^
*p* < 0.05, ^^^^
*p* < 0.01).

**Figure 5 ijms-25-01088-f005:**
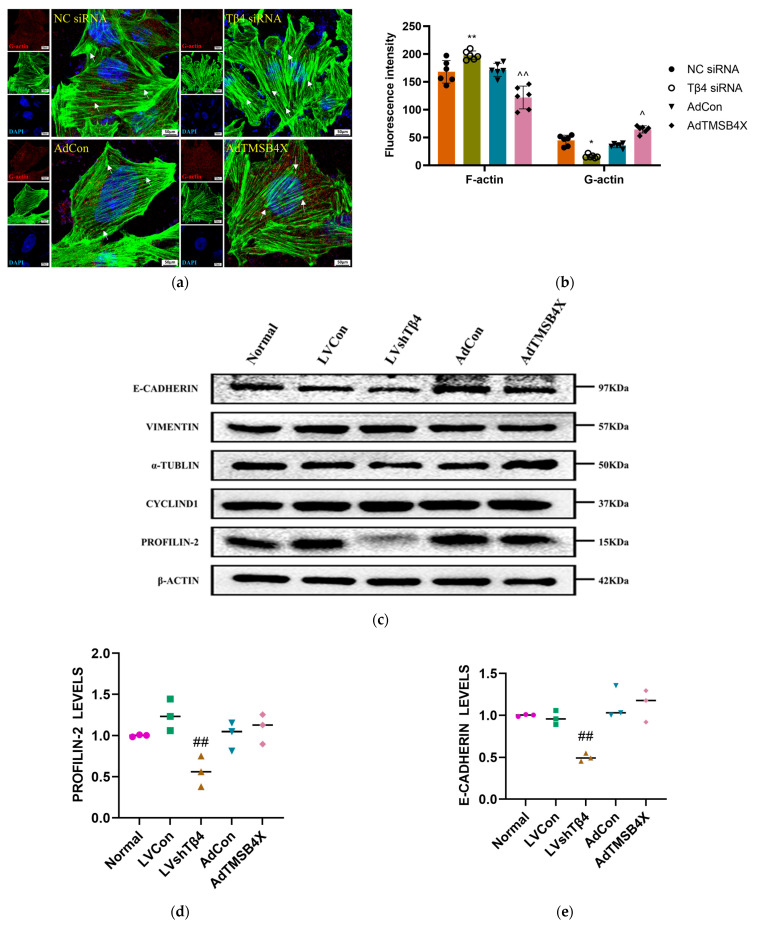
TEC cytoskeleton rearrangement and mitochondrial transfer. (**a**) Fluorescently labeled TECs from each group, treated with adenovirus and siRNA, were imaged under a 63 × 1.4 oil microscope using laser scanning confocal microscope. G-actin (red) and F-actin (green) were visualized, and nuclei were stained with DAPI (blue). White arrows indicate adhesive spots. Scale bar = 50 μm; (**b**) Statistical results of F-actin and G-actin fluorescence intensity from (**a**); (**c**) Representative Western Blot images of β-ACTIN, PROFILIN-2, CYCLIND1, α-TUBLIN, VIMENTIN, and E-CADHERIN; (**d**) Statistical results of PROFILIN-2 levels; (**e**) Statistical results of E-CADHERIN levels. Protein levels were normalized using the corresponding β-ACTIN level; (**f**) Statistical results of mRNA levels of *profilin-2*. mRNA levels were normalized using *GAPDH* by the 2^−ΔΔCt^ method; (**g**) Statistical results of mRNA levels of *e-cadherin*. mRNA levels were normalized using *GAPDH* by the 2^−ΔΔCt^ method; (**h**) Representative images of TECs mitochondria (orange), thymocytes mitochondria (purple), F-actin (green), and nuclei (blue). Representative images of enlarge selected areas in the blue and yellow boxes. Scale bar = 10 μm (bigger), 5 μm (smaller). One-way ordinary ANOVA test was used in (**d**–**g**). A two-way ANOVA test was used in (**b**). Each symbol represents a separate data point. Data for (**b**,**d**–**g**) are presented as mean ± SEM from at least three independent experiments. Statistical significance was determined by comparing the NC siRNA group (* *p* < 0.05, ** *p* < 0.01), the AdCon group (^^^
*p* < 0.05, ^^^^
*p* < 0.01), and the LVCon group (^##^
*p* < 0.01).

## Data Availability

Data is contained within the article and Appendix A.

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
