# Peer review of "Thymosin β4 Regulates the Differentiation of Thymocytes by Controlling the Cytoskeletal Rearrangement and Mitochondrial Transfer of Thymus Epithelial Cells"

_ijms, 2024, doi:10.3390/ijms25021088_

Round 1
Reviewer 1 Report (Previous Reviewer 1)
Comments and Suggestions for Authors
Dear Sirs,
The authors manly answered to all my questions and the manuscript was improved. I think that now the manuscript is suitable for publication.
Before that authors should review the text since it has many misspellings errors
Ex:
Line 361 Interestingly, Our findings were consistent with
Line 404 that Tβ4 regulateed the polymerization
The sentence
Line 345-“The two forms of G-actin/F-actin are converted to each other under particular physiological conditions. This phenomenon, known as "Tread Milling", exists simultaneously in the positive and negative microfilament assembly[22]. “
Should be replaced by
“The two forms of G-actin/F-actin are converted to each other under particular physiological conditions. This phenomenon, known as "Tread Milling", exists simultaneously in the positive and negative ends during microfilament assembly[22]. “
Author Response
Dear Sir/Madam,
Thank you very much for your letter. It is highly appreciated for the positive and constructive comments and suggestions of you and the reviewers to our manuscript entitled "Thymosin β4 regulates the differentiation of thymocytes by controlling the cytoskeletal rearrangement of thymus epithelial cells". Based on the comments, we made extensive modification on the original manuscript and added required experiments. Here, we attached revised manuscript in the formats of word for your approval, the revised part was marked with red. All authors have approved the revision of the manuscript. The response to every comments from the reviewers was listed by item as followed.
With best regards.
Yours faithfully.
1、Line 361 Interestingly, Our findings were consistent with
Response: We had modified the expression of this part to "Interestingly, our findings align with theirs."
2、Line 404 that Tβ4 regulateed the polymerization
Response: We had modified the expression of this part to "our study demonstrated that Tβ4 regulats the polymerization"
3、Line 345-“The two forms of G-actin/F-actin are converted to each other under particular physiological conditions. This phenomenon, known as "Tread Milling", exists simultaneously in the positive and negative microfilament assembly[22]. “Should be replaced by“The two forms of G-actin/F-actin are converted to each other under particular physiological conditions. This phenomenon, known as "Tread Milling", exists simultaneously in the positive and negative ends during microfilament assembly[22]. “
Response: We had modified the expression of this part to "The two forms of G-actin/F-actin undergo conversion under specific physiological conditions, a phenomenon, known as "Tread Milling", which occurs concurrently in the positive and negative microfilament assembly[22]."
In addition to fixing the grammatical errors mentioned above, we also had asked native English experts to check and correct the grammatical errors of our manuscript.
Reviewer 2 Report (New Reviewer)
Comments and Suggestions for Authors
The manuscript entitled “Thymosin β4 regulates the differentiation of thymocytes by 2 controlling the cytoskeletal rearrangement and mitochondrial 3 transfer of thymus epithelial cells” provides the data on overexpression and knock-down of thymosin beta 4 obtained by RTqPCR instead of western blot method. Without western blot on protein expression under theses condition, it is hard to clearly assess if the protein level was significantly up or downregulated. Due to this flaw, the overall quality of the manuscript is low.
Lines 36-39: recommended to split the long one sentence into two.
Lines 44-46: the sentence is unclear in terms of a stepwise differentiation of hematopoietic stem cells. In addition, the past tense used here should be changed into the present tense.
Lines 47-49: the same issue indicated right above.
Lines 40-71: the current information provided here is too complex to follow upon reading. This section should be quite shortened. Provide a scientific gap that needs to be filled by research.
Line 96-99: the latter part of the sentence seems to need a subject.
Figure 1A: the use of yellow label in the bright field images is not optimal for reading, requiring a change of color for the label.
Figure 1C: RTqPCR may not provide sufficient evidence that Tymosin Beta 4 protein expression was decreased. It is strongly recommended to provide a western blot image of the expression of the protein in each condition in Figure 1C. without knowing the protein expression levels precisely, one can mispresent the role of the protein on iTEC proliferation.
Lines 150-154: the concept of max cross-section reduction is not clear. If the main purpose of the figures 2d-2e is to illustrate the % reduction of size of thymus, the authors must present the data in a clearly and understandable manner.
Lines 1590160: appears to have a grammatical error.
Lines 162-163: a little more background on explaining CK5 and CK8.
Lines 163-166: this section was poorly explained while the information in figure 3a seems a lot to be well presented in the main text.
Lines 167-177 + Figure 3b: the choice of RTqPCR to measure the indicated mRNAs doe does not appear to be best choices. Western blot will be more powerful choice.
Lines 204-212: poor explanation on the rest of Figure 3.
Lines 234-243: the rationale for the flow cytometry is not sufficient. The use of three different markers was not explained.
Lines 262-264: this one sentence conclusion is not sufficient.
Line 304: change skeleton into cytoskeleton.
Lines 321-327: under both conditions of downregulation and upregulation of thymosin beta 4, profilin and cadherin levels decreased. A clear explanation is required.
Comments on the Quality of English LanguageIt is highly recommended to have a native speaker read the manuscript to correct grammatical errors.
Author Response
Dear Sir/Madam,
Thank you very much for your letter. It is highly appreciated for the positive and constructive comments and suggestions of you and the reviewers to our manuscript entitled "Thymosin β4 regulates the differentiation of thymocytes by controlling the cytoskeletal rearrangement of thymus epithelial cells". Based on the comments, we made extensive modification on the original manuscript and added required experiments. Here, we attached revised manuscript in the formats of word for your approval, the revised part was marked with red. All authors have approved the revision of the manuscript. The response to every comments from the reviewers was listed by item as followed.
With best regards.
Yours faithfully.
Response to the comments
1、Lines 36-39: recommended to split the long one sentence into two.
Response: We had modified the expression of this part to "Consequently, international research in aging and geriatrics has focused on addressing the question of “How to inhibit thymus atrophy?” Despite numerous research achievements, there is currently a lack of significant breakthroughs in health medicine and longevity research. "
2、Lines 44-46: the sentence is unclear in terms of a stepwise differentiation of hematopoietic stem cells. In addition, the past tense used here should be changed into the present tense.
Response: We had modified the expression of this part to "Thymocytes in the cortex originate from hematopoietic stem cells, undergoing proliferation and differentiation into small lymphocytes from the shallow to deep layers. "
3、Lines 47-49: the same issue indicated right above.
Response: We had modified the expression of this part to "T lymphocytes development and selection are not cell-autonomous processes, requiring continuous input from various cells within thymic microenvironments[8]. Chemokines secreted by stromal cells, including cTECs and mTECs, guide thymocytes through the thymus from the medullary to cortical regions and back before their egress to the periphery[3]. "
4、Lines 40-71: the current information provided here is too complex to follow upon reading. This section should be quite shortened. Provide a scientific gap that needs to be filled by research.
Response: It has been revised according to the suggestions of reviewer, because the length is too long, so the revised content can be found in the revised manuscript.
5、Line 96-99: the latter part of the sentence seems to need a subject.
Response: We had modified the expression of this part to "Therefore, this study focuses on exploring the impact of Tβ4 on the structure and spatial distribution of TECs, aiming to initially reveal the potential mechanisms of Tβ4 regulation on the differentiation and development of thymocytes. "
6、Figure 1A: the use of yellow label in the bright field images is not optimal for reading, requiring a change of color for the label.
Response: We had put labels outside the pictures to make them more visible.
7、Figure 1C: RTqPCR may not provide sufficient evidence that Tymosin Beta 4 protein expression was decreased. It is strongly recommended to provide a western blot image of the expression of the protein in each condition in Figure 1C. without knowing the protein expression levels precisely, one can mispresent the role of the protein on iTEC proliferation.
Response: We had employed Western Blot to assess Tβ4 levels in each group and added this part of the results in revised manuscript.
8、Lines 150-154: the concept of max cross-section reduction is not clear. If the main purpose of the figures 2d-2e is to illustrate the % reduction of size of thymus, the authors must present the data in a clearly and understandable manner.
Response: Because the thymus is a three-dimensional organ, and only its plane state can be photographed under the stereomicroscope, but it is its maximum cross-section, so the “maximum cross-section “was described in the manuscript. Figure 2d was different from Figure 2e. Figure 2d showed changes in the size of the thymus, while Figure 2e showed changes in the weight of the thymus.
9、Lines 159-160: appears to have a grammatical error.
Response: We had modified the expression of this part to "These findings suggest that Tβ4 may play a role in maintaining the normal morphology and size of the thymus by influencing the overflow of cortical thymocytes. "
10、Lines 162-163: a little more background on explaining CK5 and CK8.
Response: We have added an introduction about CK5 and CK8 to revised manuscript.
11、Lines 163-166: this section was poorly explained while the information in figure 3a seems a lot to be well presented in the main text.
Response:We had added an analysis of the results of this part in the revised manuscript.
12、Lines 167-177 + Figure 3b: the choice of RTqPCR to measure the indicated mRNAs doe does not appear to be best choices. Western blot will be more powerful choice.
Response: Thanks for the reviewer's advice, because this manuscript is still in the preliminary exploration stage, western blot was not used for detection. Our further study will use Western blot for detection.
13、Lines 204-212: poor explanation on the rest of Figure 3.
Response:We had added an analysis of the results of this part in the revised manuscript.
14、Lines 234-243: the rationale for the flow cytometry is not sufficient. The use of three different markers was not explained.
Response: We had added a supplement to this part in the revised manuscript.
15、Lines 262-264: this one sentence conclusion is not sufficient.
Response: We had added a supplement to this part in the revised manuscript.
16、Line 304: change skeleton into cytoskeleton.
Response: We had modified this part to "TEC cytoskeleton rearrangement and mitochondrial transfer. "
17、Lines 321-327: under both conditions of downregulation and upregulation of thymosin beta 4, profilin and cadherin levels decreased. A clear explanation is required.
Response:We had added an analysis of the results of this part in the revised manuscript.
18、Comments on the Quality of English Language
It is highly recommended to have a native speaker read the manuscript to correct grammatical errors.
Response: We had asked native English experts to check and correct the grammatical errors of our manuscript.
In addition to the reviewer’s advice, we modified Figure 3c-d, we put labels outside the pictures to make them more visible. Also, we had asked native English experts to check and correct the grammatical errors of our manuscript. Attached is our certificate of editing.

Round 2
Reviewer 2 Report (New Reviewer)
Comments and Suggestions for Authors
I would want to see Western blot results, but the authors have not done it but resubmitted it with minor changes. However, the overall data presented in the study are convincing even without Western blot results, and therefore, I am comfortable accepting the MS for publication.
This manuscript is a resubmission of an earlier submission. The following is a list of the peer review reports and author responses from that submission.
Round 1
Reviewer 1 Report
Comments and Suggestions for Authors
Dear Sirs,
The manuscript entitled “Thymosin β4 regulates the differentiation of thymocytes by controlling the cytoskeletal rearrangement of thymus epithelial cells” by Ying et al. aims to study the role of Thymosin β4 (Tβ4) in the maintenance of spatial stability of thymic epithelial cells (TECs) during aging thymus shrinkage by regulating actin polymerization/depolymerization. For this purpose, they have constructed several stable cell lines (iTECs) using lentivirus and adenovirus that constitutively express siRNAs specific for Tβ4 and overexpress Tβ4, and respective cell lines controls. They also treated iTECs with control siRNA and siRNAs specific for Tβ4. They have characterized and validated these cell lines relatively to the Tβ4 mRNA levels and showed that cell lines expressing siRNA specific for Tβ4, and cells treated with Tβ4 siRNA show decreased Tβ4 mRNA levels in comparison to controls. As expected, those overexpressing Tβ4 present Tβ4 mRNA levels higher that those found in controls. Then, the growth media of these cell lines were added to in vitro culture two-week-old mice thymus. From these experiments the authors concluded that Tβ4 levels are required to maintain normal morphology and size of thymus by inhibiting the overflow of thymocytes in thymus cortex. They pursued their studies by analyzing the effect of Tβ4 levels on the differentiation of thymus stellate epithelial cells and reticular epithelial cells. These studies were performed by immunofluorescence microscopy (IF) and RT-PCR evaluating the protein localization and mRNA levels, respectively, of key molecules involved in cell-cell adhesion, transcription regulation and receptor factors involved in cell migration. In vivo, they also directly injected the lentiviruses and adenovirus, used to construct the stable lines, in mice thymus and analyzed by electron microscopy the tissue organization. They concluded that Tβ4 affects the differentiation and spatial distribution of TECs. To assess if Tβ4 affects differentiation and development of thymocytes they analyzed the cultured thymus in presence of growth media of the stable lines above described by flowcytometry. The flowcytometry analysis supported the idea that Tβ4 affects thymocytes differentiation and development. Finally, they have investigated if Tβ4 TEC affect actin cytoskeleton by IF and by western blot they analyzed the levels of proteins involved in actin polymerization control, cell adhesion and intermediate filaments. These experiments led them to conclude that Tβ4 rearranges TEC cytoskeleton.
Although the main aim of the manuscript is pertinent, the general organization of the manuscript, the way how results are written and described, as well as the quality of some figures and results raises important concerns about the main conclusions obtained from the results. Therefore, in the present form this reviewer feels that the manuscript is not suitable for publication in Int. Journal of Molecular Sciences and requires a profound revision.
Major concerns:
-Introduction some information about the tissue complexity and architecture is missing in the introduction. This information could help those that are not familiarized with the architecture of this organ.
-Concerning the organization of the manuscript:
The authors start to describe experiments where they used the growth media from the different stable cell lines to maintain in vitro mice isolated thymus. No motivation or explanation for this approach is given. Then they inject mice directly with the constructed lentivirus and adenovirus used to create the stable cell lines (electron microscopy). This was followed by analysis of TECs directly treated with the same lentivirus and adenovirus. Therefore, three different strategies were used. How the different approaches affect the results and their coherence? The authors should explain the criteria for inclusion of different approaches putting the different results in context.
FIGURE 1.
- In graphic of b) EXPRESSION should be replaced by mRNA LEVELS.
-In c) is not shown any quantification of the cell proliferation and the image is poor.
FIGURE 2.
-As already mentioned, the authors do not explain the motivation of the experiments presented in Fig. 2 where they added growth culture media from the stable lines to the two-week-old mice thymus. Do they want to study bystander effects? This should be clarified.
-Is the observed overflow of thymocytes in cortex already described to occur in vivo? Or is a response to cell death occurring in the cultured organ overtime? Could this be an effect of growth media exhaustion?
-The conclusion that normal Tβ4 mRNA levels inhibit overflow of thymocytes in cortex is abusive because we are only adding the growth media from cells depleted of Tβ4 mRNAs. This can only be concluded analyzing the levels of Tβ4 protein in the respective cells of cultured thymus. Are the authors assuming that cells expressing siRNAs and having Tβ4 depletion do exocytosis of specific factors to the growth media that in turn are going to lower levels of Tβ4 mRNA in the cells of the thymus?
-In figure 2d the quality of the images is too poor to allow any analysis.
FIGURE 3
-The number of thymocytes showed in graphics is the total number of these cells in a complete section of the organ, or the authors defined a specific area for each of them? This should be clarified.
- The figures of the small areas are too small.
- In legend “All data were aggregated from at least three independent experiments and expressed as mean ± SEM” this sentence should be removed taking into account the graphic types presented.
FIGURE 4
-The quantification of the data of figure (4a) is missing. Is not enough to state that “Compared with LVCon group, the number of thymocytes and reticular epithelial cells in the medulla of the LVshTβ4 group were significantly reduced”. Moreover, the authors should clarify this result in comparison with that in figure 3b where they refer “However, here was no significant difference in the thymocytes number of medullary thymus Among All Groups (Figure 3b).”
-The figure 4a has poor resolution and the labels are not visible.
-In figure 4b the mRNAs were extracted from the cells of the whole organ. Is this the best way to conclude what is going on about thymocytes?
This analysis would be more correct by evaluating the protein levels by western blot.
In the legend of the figure the authors stated “Each symbol representing a separate data” – we cannot observe the symbols in the figure 4b.
Concerning figure 4c the authors mentioned that “The result revealed that thymocytes in the LVCon and AdCon groups displayed smooth cell membranes, tight intercellular connections, and normal mitochondrial morphology”. The mitochondria morphology can no be observed from this too small figures and arrows should indicate their localization. Then the figure does not allow to conclude about the morphological alterations of mitochondria in thymocytes with low Tβ4 mRNA levels.
FIGURE 5
Figure 5b- the individual points distribution cannot be observed because the graphic is too small.
FIGURE 6
- The authors state that “Changes in pericellular microfilament formation and desmosome formation in epithelial cells.”- this sentence does not make sense in the text.
-The authors state that “The morphology and complete skeleton structure of TECs, changes in pericellular microfilament formation and desmosomes formation in epithelial cells were observed by laser scanning confocal microscope”. I do not understand this sentence because in panel 6a we only can observe F-actin (actin cytoskeleton; in green) and G-actin (monomeric actin) in red and DNA stained by DAPI (blue). Therefore, specific stain for components of the desmosome structure was never used. This should be clarified.
Also, there is no analysis of G-actin staining (difficult to observe) in the different situations. The quality of the images should be improved.
- The decrease of cadherin levels, quantified from two western blots that were presented (the third one is quite difficult to analyze), seems to be smaller than that indicated in the graphics of figure 6d.
-The decrease of profilin-2 is described but not discussed in the context of actin cytoskeleton alterations described in Figure 6A, and in the context of the other analyzed protein levels. This should be improved.
DISCUSSION
Line 320 and 321- After Tβ4 was knocked down, the tightness between the thymus cells decreased. In contrast, when Tβ4 was overexpressed, thymocytes showed stronger adhesion. Please clarified this sentence in discussion since as far as I understood there no assays for cell adhesion and therefore is an indirect assumption that was not clearly tested.
MINOR CONCERNS
-In several figure legends is indicated that *P<0.05 , but in the graphics is only indicated **. Please correct this.
In figure 1-Also is mentioned “All data were aggregated (…) and expressed as mean ± SEM”. All is not adequate since this sentence is only for b), the others are not quantified.
Line 298-“Cell structure, motor morphology” should be replaced by “cell morphology during migration”
Line 299- “involved in shaping the shape of cells and maintaining cell stability” improve this sentence.
Line 305 “Tβ4, as an actin-sequestering protein, plays an important role in the regulation of G actin/F-actin balance” please insert a reference because this is known from the literature.
Reviewer 2 Report
Comments and Suggestions for Authors
In this paper, Ying and co-workers aim to study the role of thymosin B4 in the thymic role and function of thymic epithelial cells. The study is very strange in planning, with experiments that are not logically planned and have plenty of issues. The last part of the paper is more logical; However, alone it does not represent a sufficient advance. In summary, this is a lackluster effort with strange planning, hardly reproducible experimental conditions and should not be accepted under any form.
SPECIFIC ISSUES
The manner in which the paper is written indicates that the authors downregulate, or overexpress, thymosin B4, in immortalized TECs, then use the supernatant of these cells for incubation on excised thymic tissue, to study how the supernatants affect the morphology of the thymus, the TECs present in the tissue and the thymocytes. This has to be the weirdest experimental setup regarding a cytoskeletal protein. What these experiments have the potential to reveal is whether thymosin B4 affects the iTEC secretome; but this is not addressed in the paper, rendering these observations phenomenological at best. To this reviewer’s knowledge, thymosin B4 is not specifically secreted, so the effects the authors see are due to the effect of thymosin B4 depletion, or overexpression, on the secreted proteins, which then go to have an effect on thymus architecture, etc.
In a similarly ill-conceived set of experiments, the authors inject lentivirus directly into the thymus of the mice, and overinterpret their observations by suggesting that the effects seen are due to the effect of thymosin B4 depletion/overexpression on TECs in vivo. First, this crude approach would affect all present cellular types, including thymocytes themselves, so there’s no possible interpretation to these experiments except that thymosin B4 is important for thymic development in some cell type, not determined.
Finally, the authors set up to study the effect of thymosin B4 perturbation directly on the iTEC that they used to collect supernatants. The effects are subtle, but they see a decrease in profilin-2 and E-cadherin caused by depletion. However, this is clearly insufficient and the exploration of this defect, which may be more interesting, is underdeveloped.
Finally, the paper is poorly written, lack multiple references and has numerous (too many to cite them all) typos, making reading the paper a cumbersome task.
Comments on the Quality of English LanguagePoor quality of English and communication of ideas. Needs extensive replanning and rewriting.